# Improvement in Muscle Fatty Acid Bioavailability and Volatile Flavor in Tilapia by Dietary α-Linolenic Acid Nutrition Strategy

**DOI:** 10.3390/foods13071005

**Published:** 2024-03-26

**Authors:** Fang Chen, Yuhui He, Xinyi Li, Hangbo Zhu, Yuanyou Li, Dizhi Xie

**Affiliations:** 1College of Marine Sciences of South China Agricultural University, Guangzhou 510642, China; chenfang@scau.edu.cn (F.C.); heyuhui@scau.edu.cn (Y.H.); xinyili@stu.scau.edu.cn (X.L.); m200100293@st.shou.edu.cn (H.Z.); yyli16@scau.edu.cn (Y.L.); 2Maoming Branch, Guangdong Laboratory for Lingnan Modern Agriculture, Guangzhou 510642, China; 3Marine Biology Institute & Guangdong Provincial Key Laboratory of Marine Biotechnology, Shantou University, Shantou 515063, China

**Keywords:** farmed tilapia, α-linolenic acid, fatty acid distribution, volatile odor, muscle quality

## Abstract

To investigate the modification of muscle quality of farmed tilapia through dietary fatty acid strategies, two diets were formulated. Diet SO, using soybean oil as the lipid source, and diet BO, using blended soybean and linseed oils, each including 0.58% and 1.35% α-linolenic acid (ALA), respectively, were formulated to feed juvenile tilapia for 10 weeks. The muscular nutrition composition, positional distribution of fatty acid in triglycerides (TAGs) and phospholipids (PLs), volatile flavor, lipid mobilization and oxidation were then analyzed. The results showed that there was no distinct difference between the SO and BO groups in terms of the nutrition composition, including crude protein, crude lipid, TAGs, PLs, and amino acid. Although the fatty acid distribution characteristics in ATGs and PLs showed a similar trend in the two groups, a higher level of n-3 PUFA (polyunsaturated fatty acid) and n-3 LC-PUFA (long-chain polyunsaturated fatty acid) bound to the glycerol backbone of TAGs and PLs was detected in the BO group than the SO group, whereas the opposite was true for n-6 PUFA. Additionally, the muscular volatile aldehyde and alcohol levels were higher in the BO group. Moreover, the expression of enzymatic genes and protein activities related to lipid mobilization (LPL, LPCAT, DGAT) and oxidation (LOX and GPX) was higher in the BO group. The results demonstrate that high-ALA diets may improve the fatty acid bioavailability and volatile flavor of tilapia by improving the lipid mobilization and oxidation, which provides new ideas for the improvement of muscle quality in farmed fish.

## 1. Introduction

Livestock, poultry, and aquatic products, which are abundant in protein, vitamins, and minerals, are widespread components of human diets. Compared to livestock and poultry products, aquatic products provide more nutrient options, such as n-3 long-chain polyunsaturated fatty acids (n-3 LC-PUFAs), specifically, eicosapentaenoic acid (EPA; 20:5n-3) and docosahexaenoic acid (DHA; 22:6n-3) [1,2], which play fundamental roles in reducing inflammatory and cardiovascular disease, as well as promoting growth and development [3]. While capture-fisheries production has been stagnant for decades, consumers have resorted to farmed fish products due to increased requirements for n-3 LC-PUFA [4]. The aquatic fish species tilapia (an *Oreochromis* sp.) is widely cultivated in the world due to its high productivity, disease resistance, wide adaptability and low feed input [5,6], with annual production up to 7.3 million tons in 2021 [7]. However, cultured tilapia usually receive soybean oil- and plant protein-supplemented diets (n-3 LC-PUFA-free), which decreases the quality of the farmed products [8]. Although somewhat controversial, farmed tilapia are labeled as low-class aquatic products, probably due to their poor muscle nutrition. The production of high-quality tilapia would be a promising way to address that issue in tilapia culture.

Previous studies showed that tilapia possess a high capacity for converting linoleic acid (LA, 18:2n-6) and α-linolenic acid (ALA, 18:3n-3) into n-6 and n-3 LC-PUFA, respectively [9,10], and diets supplemented with 0.45–0.70% ALA are beneficial for fish growth [11]. Additionally, muscle fatty acid nutrition was improved in farmed tilapia by feeding them linseed oil diets (contained 1.35% ALA) and fish oil diets (enriched in n-3 LC-PUFA) [10,12,13], revealing that the muscular fatty acid nutrition of tilapia can be modified through a dietary lipid and fatty acid nutrition strategy. In addition to the n-3 LC-PUFA levels, the bioavailability of beneficial fatty acids also determines the nutritional value for consumers [14,15]. Dietary n-3 LC-PUFAs are mainly present in the form of triglyceride (TAGs) or phospholipids (PLs), and most of the evidence indicates that n-3 LC-PUFA in the PL form has better bioavailability [15,16]. Furthermore, the fatty acid distribution in the glycerol backbone of TAGs and PLs has a marked impact on their digestibility, metabolism, and the bioavailability of fatty acids [14,17]. Studies on mammals showed that sn-2 esterified saturated fatty acid (SFA) is more efficiently absorbed than that at the sn-1 and sn-3 (sn-1/3) positions, which are prone to forming insoluble soaps with ions [17]. Liu et al. [18] reported that palmitic acid is mainly located at the sn-2 positions of TAGs in tilapia products, while unsaturated fatty acids (UFAs) are mainly positioned at sn-1/3 positions, which is beneficial to absorption and utilization for consumers. Studies on aquatic animals demonstrated that muscle fatty acid distribution is associated with dietary sources of lipids [19,20], while its potential mechanism remains unclear.

In addition, compared to monounsaturated fatty acids (MUFAs) and SFA possessing fewer unsaturated double bonds, LC-PUFA is more susceptible to oxidation [21]. Oxidation products derived from fatty acids through enzymatic oxidation (catalyzed by lipolysis and lipoxygenase) and autoxidation are dominant factors in the formation of volatile compounds, which contribute to the flavor quality of fish products [22]. Although few studies on aquatic animals have indicated that muscle volatile compounds are influenced by dietary lipid sources and fatty acids [23,24], the interrelationships among dietary fatty acids, muscle n-3 LC-PUFA bioavailability, and volatile compounds have rarely been documented in fish. Therefore, the mechanisms and consequences of high-ALA diets on muscle fatty acid nutrition and bioavailability, as well as volatile compounds in tilapia, are investigated in this study. The results will provide new information for improving the nutritional quality of cultivated tilapia through a fatty acid nutritional strategy.

## 2. Materials and Methods

### 2.1. Tilapia Feeding Trial

To investigate the modification of muscle-beneficial fatty acid bioavailability and volatile compounds in tilapia through dietary ALA strategies, muscle samples from our previous feeding trial were used [13]. In the feeding trial, two diets with different ALA levels were formulated (the specific dietary ingredients are shown in Appendix A), namely diets SO (using soybean oil as the lipid source) and BO (using blended soybean and linseed oils as the lipid source) with 0.58% and 1.35% ALA levels, respectively. In the feeding trial, 120 tilapia GIFI (initial body weight of 170 g) were randomly distributed into 6 tanks (300 L, 20 fish per tank and each dietary group with three tanks), and cultured for 10 weeks with the SO or BO diets. The detailed management of fish husbandry was described in our pervious study [13]. After the feeding experiment, tilapia (final weight ~400 g) were anesthetized with phenoxyethanol prior to weighing and sampling. Muscle samples were randomly collected from six fish per tank to determine the proximate composition. Another six samples were randomly gathered from each tank, and the isolated samples were frozen immediately in liquid nitrogen, then stored at −80 °C until analysis. The fish-rearing protocols and sample handling in this study were approved by the Animal Care and Use Committee of South China Agricultural University (SCAU-AEC-2010-0416).

### 2.2. Analysis of Muscle Proximate Compositions

The muscular proximal composition, including moisture, ash, crude lipid, and crude protein were analyzed as previously described [10]. In short, the moisture content was analyzed by drying samples to constant weight at 80 °C. The ash content was measured by incinerating the dried samples at 550 °C for 6 h. The crude lipid content was extracted with petroleum ether by Soxhlet extraction, and the crude protein content was determined using the Kjeldahl method.

### 2.3. Evaluation of Muscle Amino Acid Compositions

According to the description of the previous study [25], 50 mg muscle samples were hydrolyzed with 6 M hydrochloric acid at 110 °C for 22–24 h, then the amino acid composition were measured by using an Amino Acid Analysis System (Hitachi L-8900, Wako, Japan). The contents of the amino acid were expressed as grams per 100 g protein of the sample.

### 2.4. Evaluation of Muscle Triglycerides and Phospholipids Contents

According to our previous study [10], the muscle lipid was extracted with chloroform/methanol (2:1; *v*/*v*). Separation of muscle TAG and PL was performed by the thin layer chromatography (TLC) method. Briefly, total lipids (~0.1 g) were detected on 200 mm × 200 mm Silica Gel TLC plates (Haiyang Chemical Group, Qingdao, China). Firstly, PL was spread in a solvent system with 0.25% KCl/methanol/isopropyl acid chloroform/methyl acetate (9/10/25/25, *v*/*v*/*v*/*v*) for approximately 100 min. Subsequently, TAG was separated in another solvent system with acetic acid/ether/isohexane (1.5/25/80, *v*/*v*/*v*) for 160 min. The PL and TAG bands were extracted using the chloroform and methanol method after iodine staining, respectively.

### 2.5. Analysis of Fatty Acid Distribution in TAG and PL

The fatty acid distribution of TAG and PL was determined by a slightly modified method of enzymatic hydrolysis [26]. Briefly, purified TAG was dissolved into 0.05% bile salts and 1 M tris-HCl solution (pH 7.6), then hydrolyzed with pancreatic lipase solution (10 mg pancreatic lipase in 0.1 mL 2.2% CaCl_2_). The TAG hydrolysate was separated by an NH2 SPE cartridge (ANPEL Laboratory Technologies Inc., Shanghai, China), then washed with ethyl acetate/hexane (15/85, *v*/*v*, 8 mL), and sn-2-MG was eluted with methanol/methylene chloride (1/2, *v*/*v*, 6 mL), to determine the fatty acid composition.

Purified PL were hydrolyzed by 50 μg/mL phospholipase A2 solution (in 1 M, pH 8.0 tris-HCl buffer), then the hydrolysate was separated by an NH2 SPE cartridge, and the Lyso-PLs and FFAs were eluted by 5 mL methanol and 5 mL ether/acetic acid (98/2, *v*/*v*), respectively, and to analyze the fatty acid composition by concentrating with N2.

### 2.6. Analysis of Fatty Acid Compositions

As in the previous study [10], the fatty acid composition of purified TAG, PL, sn-2 MAG, FFAs and Lyso-PLs were determined. In brief, the fatty acid methyl esters (FAME) of the above lipid aliquots were produced using boron trifluoride, the fatty acid composition was determined using gas chromatography (Shimadzu emit Co., Ltd., Tokyo, Japan) by comparing FAME with commercial standards (Sigma, St. Louis, MO, USA). Fatty acid composition at sn-1/3-positions were calculated from their contents in the TAG and sn-2-MAG by: sn-1/3-positions % = (TAG % ×3 − sn-2-MAG %)/2.

### 2.7. Analysis of Lipid Mobilization and Oxidation in Muscle

For analysis of lipid mobilization and oxidation in the muscle, samples were weighed and homogenized with three volumes (*w*/*v*) of ice-cold buffer (pH 7.4, 0.25 M sucrose, 0.02 M Tris–HCl, 2 mM EDTA, 0.1 M sodium fluoride, 0.01 M β-mercapto-ethanol, 0.5 mM phenyl methyl sulfonyl fluoride), and centrifuged at 5000× *g* for 20 min at 4 °C. Then the collected supernatant was used for measuring the enzymatic activities of adipose triacylglyceride lipase (ATGL), lipoprotein lipase (LPL), lysophosphatidylcholine acyltransferase (LPCAT), lipoxygenases (LOX), glutathione peroxidase (GPX) and malondialdehyde (MDA) contents. The activities of ATGL, LPL, LPCAT and LOX were measured following the instructions of manufacturer (Enzyme Linked Biotechnology Co., Ltd., Shanghai, China), while GPX activities and MDA contents were detected using sampling kit purchased from Nanjing Jiancheng Bioengineering Institute (Nanjing, China).

The peroxide value (POV) of muscle was estimated following the method reported by [27]. Total muscle lipid extractions were mixed with orange xylenol (50 μL; 10 mM), and FeCl_2_ (50 μL; 18 mM) solution. The OD absorbance value of the sample was detected by spectrophotometer at 560 nm. Each sample had at least three replicates.

### 2.8. Determination of Volatile Flavor Compounds

The volatile compounds of muscle were determined by the HS-SPME-GC-MS method according to the previous study [28]. In short, a muscle sample (5.0 g) was added to a headspace bottle and maintained at 45 °C for 30 min. HS-SPME extraction was performed at 60 °C for 30 min with an SPME fiber, and the gas chromatography-mass spectrometer system (Agilent 7890A-5975C; Agilent Technologies Inc., Santa Clara, CA, USA) was used to measure the volatile compounds. The acquired volatile compounds were identified by retention index (RI) and reverse match factor (similarity > 700).

### 2.9. Analysis of Real-Time Quantitative PCR

Following the kit protocols, the total RNA of muscle samples was first extracted, and then reverse transcribed into cDNA. The relative expression of mRNA levels of genes involved in lipid mobilization (*atgl*, *lpl*, *dagt1*, *dagt2*, *lpcat3*, *lpcat4*) and fatty acid oxidation (*lox5*, *lox12*, *lox15*, *gpx1*, *gpx4*) in the muscle of tilapia was measured using quantitative real-time PCR (qPCR) with specific primers (Appendix A). The reaction procedures of qPCR were performed according to [13], and the relative gene expression levels were calculated using the 2^−∆∆CT^ method. Triplicate analyses were conducted for each sample.

### 2.10. Statistical Analysis

Data are expressed as the mean ± standard error of the mean (SEM, n = 3). One-way analysis of variance (ANOVA) and Student’s *t*-test was conducted by SPSS 22.0 (SPSS Inc., Chicago, IL, USA). *p* < 0.05 was considered significantly different. The relationships between the volatile compounds and samples from different dietary groups were performed using Hierarchical cluster analysis (HCA) by the online software ImageGP (http//:www.bic.ac.cn/ImageGP/, accessed on 14 March 2024). With the help of SIMCA 14.1 software, principal component analysis (PCA) was implemented to detect discrepancies of muscle volatile compounds between the SO and BO groups, and orthogonal partial least squares discriminant analysis (PLS-DA) was also performed to confirm key volatile compounds.

## 3. Results

### 3.1. Proximate and Amino Acid Composition of Muscle

The muscular proximate and amino acid composition are shown in Table 1 and Table 2. Comparing to the SO group, the proximate composition (such as moisture, crude protein, crude lipid, and crude ash) of muscle in BO group showed no statistical difference (*p* > 0.05). Additionally, the muscular total amino acids (TAAs), essential amino acids (EAAs), EAA/TAA ratios, flavored amino acids (FAAs), non-essential amino acids (NEAAs) and semi-essential amino acids (SEAA) also did not show significant differences between SO and BO groups (*p* > 0.05).

### 3.2. Fatty Acid Compositions of ATG and PL of Muscle

The contents of muscle ATG and PL are shown in Table 1. Though relatively higher contents of PL and ATG were measured in the BO group than the SO group, they did not reach significant differences (*p* > 0.05). As a consistent change in the ATG and PL contents was detected between the two groups, the fatty acid composition of ATG and PL showed a similar variation trend (Table 3). For example, the percentage of ALA, EPA and DHA, as well as n-3 PUFA and n-3 LC-PUFA in muscle ATG and PL of the BO group, was significantly higher than that of the SO group, while the opposite was true for LA, SFA and n-6 PUFA. The results suggest that n-3 PUFA and n-3 LC-PUFA-containing TAG and PL were characterized as BO diet-induced muscle lipids, while SFA and n-6 PUFA-containing TAG and PL were characterized as SO diet-induced muscle lipids.

### 3.3. Positional Distribution of Fatty Acids in the TAG and PL of Muscle

The positional distributions of fatty acids in TAG molecules are shown in Figure 1A and Appendix A. High proportions of 16:0, 18:0, DHA, SFA and n-3 LC-PUFA were detected in the sn-2-position, while 16:1, 18:1, LA, ALA, ARA, EPA, MUFA and n-6 PUFA were preferentially positioned at the sn-1/3 (*p* > 0.05) position. Regarding the fatty acid distribution of PL molecules (Figure 1B and Appendix A), the percentage of SFA, n-3 PUFA and n-3 LC-PUFA in the sn-2 position was higher than that in the sn-1 position (*p* < 0.05), while LA and ALA were preferentially distributed in the sn-1 position, which had the same trend as that of TAG molecules. Although the positional distribution characteristics of most fatty acids in ATG and PL showed a similar trend, high concentrations of EPA and ARA tended to accumulate at the sn-1/3 and sn-2 positions of ATG and PL molecules, respectively. Among the two groups, the amount of n-3 PUFA and n-3 LC-PUFA at the sn-2 and sn-1/3 (or sn-1) positions of TAG and PL in the BO group was much higher than that in the SO group (*p* < 0.05), whereas the opposite was true for n-6 PUFA.

### 3.4. Identification and Composition of Volatile Compounds

A heat map of volatile substances obtained by GC-MS is shown in Figure 2A and Appendix A. The distribution of different volatile substances in the two groups is different. The contents of decane, undecane and dodecane were relatively higher in the SO group, while higher 1-octen-3-ol, 1-octanol, nonanal and decanal contents were observed in the BO group. As shown in Figure 2B, the volatile compounds with oily, waxy, fruity, sweet, and floral odors were higher in the BO group than those in the SO group (*p* < 0.05), whereas the opposite was true for the compounds with pungent acrid, gasoline-like, and faint odor and lacking odor, which suggested that BO diets can modify the farmed tilapia flavor by favoring the contents of volatile aldehydes and alcohols.

Based on the first two principal components, two well-separated clusters were detected between the SO and BO groups (Figure 2C), and the cumulative contribution rate was 84.4% (PC1: 79.4%. PC2: 5.0%), which indicated that the muscle volatile compounds of the two dietary groups can be effectively distinguished by the PCA model (Figure 2C). Additionally, 13 important contributors, including 3,8-dinosethyl-decane, 2-methyl-decane, hexadecane, etc., were identified for volatile differentiation of fish-fed dietary SO and BO using identical screening criteria (VIP > 1.0, *p* < 0.05) (Figure 2D).

### 3.5. Enzyme Activity and Gene Expression Related to Lipid Mobilization

As Figure 3 shows, the enzymatic activity of LPL of the fish-fed BO diets was higher than that of the fish-fed SO diets (*p* < 0.05). Although there was no statistical difference between the two groups (*p* > 0.05), relatively higher ATGL and LPCAT3 enzymatic activities were detected in the BO group (Figure 3A). Accordingly, higher mRNA levels of genes related to lipolysis (*atgl* and *lpl*) were also observed in the muscle of fish fed BO diets (*p* < 0.05) (Figure 3C). Additionally, the mRNA levels of genes related to lipid synthesis (*dgat2*, *lpcat3*, *lpcat4*) in the BO group were higher than those of the SO group (Figure 3C). The results indicate that BO diets with high ALA levels are beneficial for lipid mobilization in tilapia muscle.

### 3.6. Fatty Acid Oxidative Stability

The oxidative stability of muscle lipids was evaluated by the enzymatic activities of LOX and GPX, their gene expression levels, and the contents of POV and MDA, which are shown in Figure 3B,D. Compared to the SO group, the muscular LOX activities in the BO group were significantly higher (*p* < 0.05). Although the POV content and GPX activities of the BO group were 34.62% and 29.60% higher than those of the SO group, respectively, the content of neither POV nor MDA nor GPX activities demonstrated a statistical difference between the SO and BO groups (*p* > 0.05). Accordingly, the mRNA levels of genes related to fatty acid oxidation (*lox5*, *lox12*, *lox15*, *gpx1*, and *gpx4*) in the BO group were higher than those in the SO group (*p* > 0.05) (Figure 3D). Specifically, muscular *lox12* and *gpx4* mRNA levels in the BO group were eleven- and fivefold those in the SO group (*p* < 0.05), respectively, which is a greater variation than those of other *lox genes* (*lox2* and *lox15*) and *gpx1* genes.

## 4. Discussion

Numerous studies have declared that the sharp decline in muscular fatty acid nutrition in farmed fish fed diets with plentiful substitutions of fish oil is a predicament in the quality of cultured fish production [29,30]. To address this issue, the feeding strategy and supplementation of omega 3-enriched plant oils (such as linseed oil, engineered plant oil, etc.) were applied to cultured fish to increase the muscular omega 3 contents, especially for n-3 LC-PUFA [24,31,32]. Previous studies on tilapia also showed that high α-linolenic acid diets improved n-3 LC-PUFA levels and muscle n-3/n-6 PUFA ratios in this fish [10,33]. Thus, increasing ALA intake is an efficient strategy to accumulate n-3 LC-PUFA levels in muscle lipid molecules and improve the nutritional quality of farmed tilapia. However, the bioavailability of n-3 LC-PUFA differs in the TAG or PL form, with higher bioavailability in the latter [15,16]. Notably, in this study, a common feature was found whereby a high percentage of n-3 LC-PUFA was consolidated into muscle TAG and PL of fish fed high dietary ALA levels, which favors the n-3 LC-PUFA bioavailability of tilapia. Additionally, relatively high lipid contents were detected in the muscles of fish fed high-ALA diets. These observations are in agreement with the idea that the intake of high dietary n-3 PUFA is conducive to the muscle lipid levels in farmed tilapia and can become comparable to those of golden pompano (*Trachinotus ovatus*), gilthead sea bream (*Sparus aurata*) and largemouth bass (*Micropterus salmoides*) [33,34,35]. The results probably indicate that such a high dietary ALA strategy is beneficial to the muscular fatty acid nutrition of the farmed tilapia by improving the esterification of n-3 LC-PUFA in the lipid molecules, such as TAG and PL.

In addition to the esterified n-3 LC-PUFA contents, the position of fatty acids on the glycerol backbone of TAG and PL also determines the bioavailability of fatty acids [14,17]. It is well known that 16:0 and 18:0 at the sn-1/3 positions are preferred for lipase and tend to form unabsorbable fatty acid soaps, while these fatty acids in the sn-2 position slowed down the clearance of chylomicron, which is beneficial for the isomerization of 2-MAG and the efficient hydrolyzation of fatty acids [17]. Similarly, in the present study, the preferential deposition of 16: 0 at the sn-2 position of muscle TAG and PL molecules in tilapia was not related to dietary ALA levels, and most UFAs were positioned at the external positions of TAG, confirming that the enrichment of palmitoyl triacylglycerols (OPO/OPL) in the lipid of tilapia is a potential human milk fat substitute [18]. However, the DHA of muscle TAG in tilapia fed SO or BO diets was mainly distributed in sn-2 positions, which is in agreement with that of other freshwater and marine fish species, as well as wild tilapia [26,36]. Compared to sn-1/3-positioned DHA, sn-2-positioned DHA in TAG is efficiently absorbed by the intestinal mucosa, which is beneficial to the brain [37]. Unlike the positional distribution of ARA and EPA in the TAG fraction, ARA, EPA and DHA tend to remain at the sn-2 position of PL, as reported in another study of tilapia, where the distribution of LC-PUFA of PL was inconsistent with that of TAG [26], which confirms that LC-PUFA is typically associated with the sn-2 position in phospholipids of freshwater and marine fish species [38,39,40].

Although the n-3 LC-PUFA distribution of lipid molecules in tilapia was impervious to dietary ALA levels, the ratio of DHA positioned in sn-2 in PL of the fish fed SO diets was higher than that of the fish fed BO diets (91.70% vs. 81.67%, in SO and BO, respectively), which is in agreement with the finding that low n-3 LC-PUFA diets seemed to retain DHA in the sn-2 position of the PL, given the lower availability and the important structural role of DHA [41]. It is well known that the bioavailability of fatty acids is dependent on the interaction of the dietary lipid structure with endogenous metabolism [14]. However, soybean and linseed oils have the same fatty acid distribution, with the unsaturated fatty acid and SFA positioned at sn-2 and sn-1/3 of lipid fractions, respectively [14,42], which is inconsistent with the fatty acid distribution of tilapia muscle. Hence, the fatty acid distribution of tilapia muscle could be caused by the reaction preference of substrates involved in the lipid biosynthesis [36]. LPCAT and ATGL play an important role in TAG and PL (only LPCAT) biosynthesis by catalyzing the rate-limiting steps [41]. For the substrate specificity of LPCAT, LPCAT3 and LPCAT4 preferentially acidylate LC-PUFA-CoA at the sn-2 position of lysophospholipid to produce phosphatidic acid [43], which can further synthesize PL and TAG. The final step of TAG synthesis is catalyzed by DGAT, where ATGL2 displayed a striking preference toward diacylglycerol containing unsaturated fatty acids [44]. In the present study, high LPCAT activity and *lpcat3*, *lpcat4*, and *atgl2* mRNA levels were detected in the tilapia fed high-ALA diets, which confirms that high n-3 PUFA diets enrich the n-3 LC-PUFA positioned at sn-2 in PL and TAG [19,41]. Hence, the results suggest that high-ALA diets are conducive to the fatty acid distribution of TAG and PL (lipid remodeling) in tilapia muscle favored for consumer utilization.

Apart from the muscular nutritional value and bioavailability of fatty acids, PUFA also contributes to the volatility and flavor of meat by lipid oxidation [21,45,46]. Considerable evidence indicates that muscle UFA, especially n-3 PUFA, plays a particularly important role in the formation of volatile and flavor compounds in livestock, poultry, and aquatic products [45,46]. The popularity of aquatic products among consumers is determined by their sensory characteristics, which are affected by volatile odor to a great extent [46]. In the present study, high concentrations of volatile aldehydes and alcohols were observed in the muscle of fish fed BO diets (Figure 2), which is consistent with the muscle n-3 PUFA and LC-PUFA levels. Volatile aldehydes and alcohols produced by the oxidative decomposition of PUFA are responsible for the pleasant taste, such as mushroom flavor, fruit flavor, tallow flavor or nut flavor [47]. Wood et al. [46] also found that n-3 PUFA oxidation products were directly responsible for the particular flavor of livestock. In this study, the muscle of fish in the BO group was rich in volatile compounds with oily, waxy, fruity, sweet, and floral odors, suggesting that high-ALA diets can modify the flavor of tilapia from the farms by improving the lipid oxidation products of volatile aldehydes and alcohols. Actually, the sensory quality of tilapia fed high-ALA diets needs to be investigated further by electronic nose systems and consumer panels.

Volatile compounds were produced from the lipid flavor compounds by a series of lipolysis and oxidation [22]. Lipase, such as LPL and ATGL, hydrolyzes the ester bonds of lipids to yield fatty acids, and LOX further converts the oxidation of fatty acids (especially for PUFA) into oxy-radicals and aldehyde [21]. In this study, high LPL, ATGL and LOX enzymatic activities, as well as their corresponding gene expression levels, were detected in the muscle of tilapia fed high-ALA diets. Specifically, muscular *lox12* and *lox15* mRNA levels in the BO group increased significantly, which is consistent with the observation of LC-PUFA oxidation (particularly ARA and EPA) initiated by LOX12 and LOX15 [48]. Compared to LOX, GPX is a key antioxidant regulator of systemic redox homeostasis, which can prevent the accumulation of toxic lipid oxy-radicals [49] and convert lipid hydroperoxide into harmless lipid alcohols [50]. Notably, high muscle *gpx4* mRNA levels were observed in the BO group, which is positively correlated with the volatile alcohol contents, indicating that high muscle volatile alcohols in the BO group may be induced by GPX4 activity [50]. The above results indicated that high levels of volatile aldehydes and alcohols in the BO group may be induced by high enzymatic lipid oxidation [22,46].

## 5. Conclusions

In summary, based on the nutritional value of muscle and bioavailability of fatty acids, volatile flavor, enzyme activity and gene expression related to lipolysis and lipid oxidation, the present study demonstrated that high-ALA diets increased n-3 LC-PUFA levels in the muscle of tilapia and modified the bioavailability of beneficial fatty acids in lipid molecules by improving lipid remodeling (LPCAT). Additionally, the muscular volatile aldehydes and alcohol levels increased with the increased muscle n-3 PUFA and LC-PUFA levels, as well as dietary ALA supply. Furthermore, high-ALA diets are beneficial for lipolysis (LPL and ATGL) and fatty acid oxidation (LOX and GPX) of farmed tilapia. These findings indicate that high α-linolenic acid diets modify muscle fatty acid bioavailability, as well as volatile odors by enhancing lipid mobilization (lipid reesterification and lipolysis) and fatty acid oxidation in farmed tilapia (Figure 4). To our knowledge, our findings provide new insights for understanding the modification of muscle nutrition and odor value in farmed tilapia by dietary fatty acid strategies.

## Figures and Tables

**Figure 1 foods-13-01005-f001:**
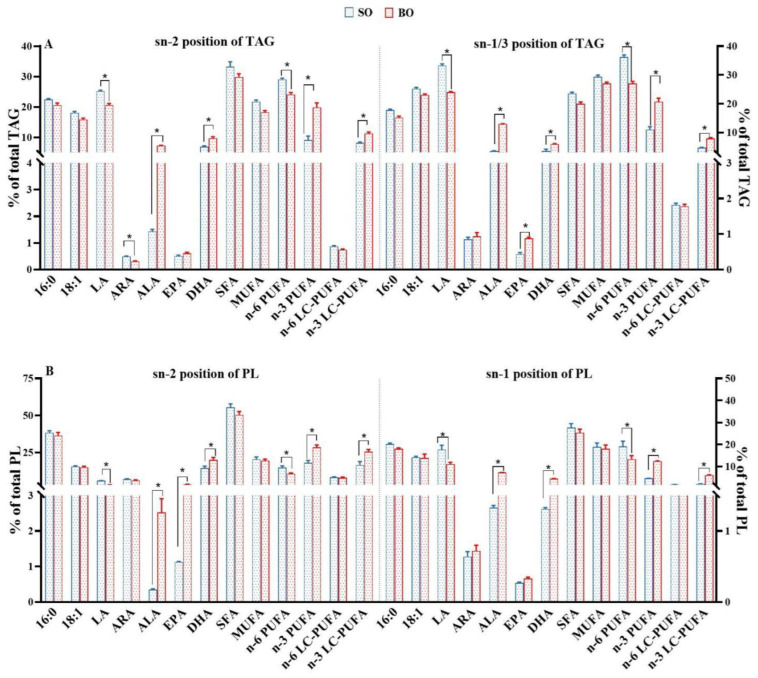
Fatty acid distribution of TAG and PL in the muscle of tilapia fed the two different diets. (**A**) Positional distribution of main fatty acids in the TAG fractions; (**B**) positional distribution of main fatty acids in the PL fractions. Values are means ± SEM (n = 3). * *p* < 0.05.

**Figure 2 foods-13-01005-f002:**
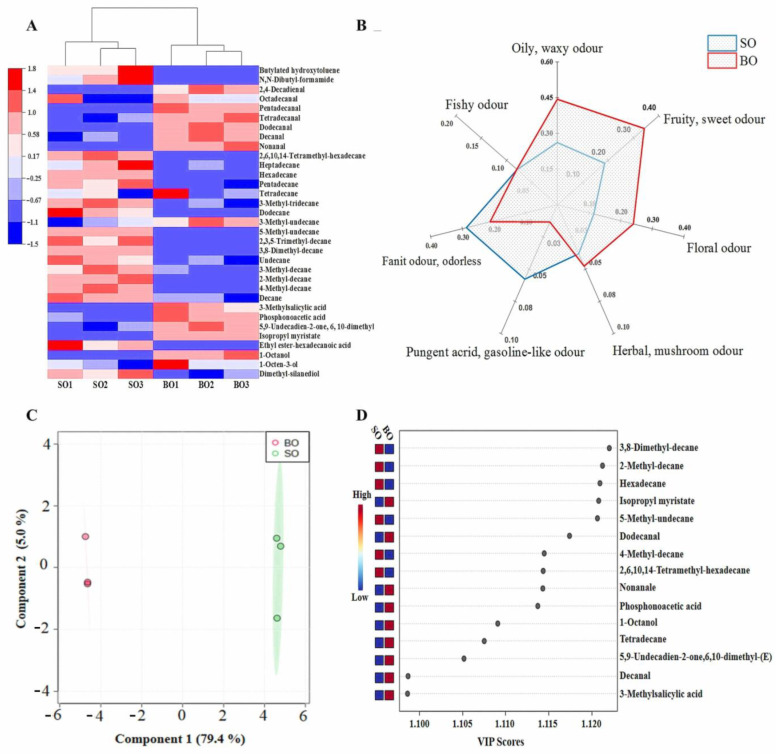
Volatile flavor compounds in the muscle of tilapia fed the two different diets. (**A**) Clustering heat map of identified volatile flavors; (**B**) volatile compound description profile; (**C**), partial square discriminant analysis (PLS-DA) of identified volatile flavors; (**D**) VIP scores of identified volatile flavors. The detailed values were listed in Appendix A.

**Figure 3 foods-13-01005-f003:**
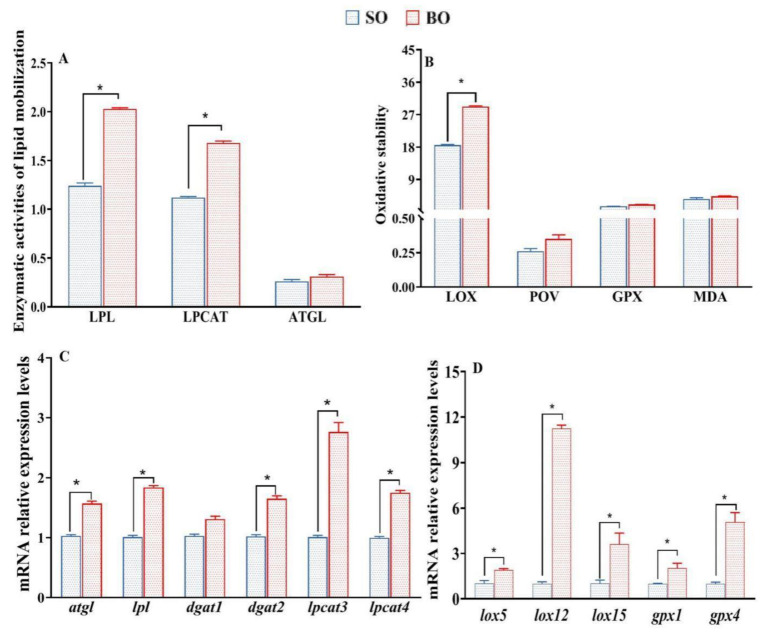
Evaluation of lipid mobilization and fatty acid oxidation in muscle of tilapia fed the two different diets. (**A**) Protein activities related to lipid mobilization (LPL, LPCAT, and ATGL are expressed as U mg prot^−1^); (**B**) evaluation of oxidation stability (LOX, POV, GPX and MDA are expressed as meq kg^−1^, U g prot^−1^, U mg prot^−1^, and nmol mg prot^−1^, respectively); (**C**), relative expression levels of genes related to lipid mobilization; (**D**) relative expression levels of genes related to fatty acid oxidation. Values are means ± SEM (n = 3). * *p* < 0.05.

**Figure 4 foods-13-01005-f004:**
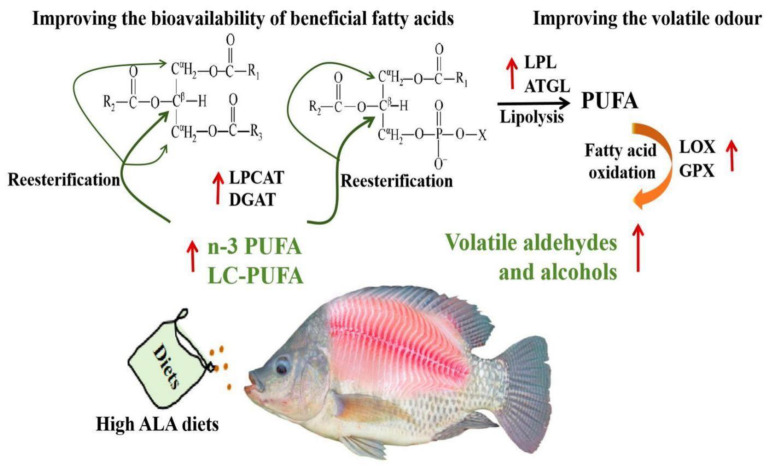
Schematic overview of improvement in muscle fatty acid bioavailability and volatile flavor in tilapia by diets high in α-linolenic acid.

**Table 1 foods-13-01005-t001:** Muscular proximate composition and lipid contents of tilapia fed the two different diets.

	Dietary Groups	
	SO	BO
Proximate composition (%)		
Moisture	77.35 ± 0.51	77.17 ± 0.95
Crude protein	88.41 ± 0.24	88.01 ± 0.25
Crude lipid	3.76 ± 0.17	4.24 ± 0.26
Crude ash	1.26 ± 0.07	1.15 ± 0.06
Lipid contents (mg g^−1^)		
TAG	9.12 ± 0.13	9.37 ± 0.19
PL	2.02 ± 0.15	2.55 ± 0.22

**Table 2 foods-13-01005-t002:** Muscular amino acid contents of tilapia fed the two different diets.

Main Amino Acid Compositions (mg g^−1^)	Dietary Groups	
	SO	BO
Lys	7.31 ± 0.69	7.21 ± 0.53
Phe	3.79 ± 0.15	3.71 ± 0.13
Met	2.21 ± 0.24	2.07 ± 0.25
Thr	3.98 ± 0.10	3.90 ± 0.08
Ile	3.82 ± 0.14	3.78 ± 0.16
Leu	6.19 ± 0.32	6.05 ± 0.25
Val	4.11 ± 0.11	4.08 ± 0.14
His	1.92 ± 0.14	2.01 ± 0.16
Arg	5.93 ± 0.21	5.87 ± 0.25
Asp	8.86 ± 0.54	8.63 ± 0.59
Ser	3.63 ± 0.13	3.57 ± 0.18
Glu	11.96 ± 0.68	11.69 ± 0.71
Gly	5.48 ± 0.11	5.47 ± 0.15
Ala	5.02 ± 0.13	5.14 ± 0.16
Cys	0.58 ± 0.04	0.54 ± 0.03
Tyr	2.10 ± 0.10	2.18 ± 0.12
Pro	3.05 ± 0.05	3.11± 0.06
FAAs	30.76 ± 2.56	28.42 ± 2.87
EAAs	31.41 ± 2.30	31.22 ± 1.95
SEAAs	7.85 ± 0.69	7.88 ± 0.75
NEAAs	40.68 ± 3.98	41.05 ± 3.74
TAAs	79.94 ± 5.88	80.15 ± 5.02
EAA/TAAs	0.39 ± 0.00	0.39 ± 0.00

Note: Values are means ± SE (n = 3) with nine fish per treatment. Tryptophane was not detected. FAA, flavored amino acid (including Phe, Asp, Gly, Ala, Thr, Ser); EAAs, essential amino acids (including Lys, Thr, Phe, Met, Val, Ile, Leu); SEAAs, semi-essential amino acids (including His, Arg); NEAAs, non-essential amino acids (including Asp, Ser, Glu, Gly, Ala, Cys, Tyr); TAAs, total amino acids.

**Table 3 foods-13-01005-t003:** Fatty acid composition of muscle TAG and PL in tilapia fed different diets.

Main Fatty Acid Compositions (%)	Dietary Groups	
	SO	BO
TAG		
16:0	19.12 ± 0.39 ^a^	15.82 ± 0.84 ^b^
18:0	6.88 ± 0.21 ^a^	5.98 ± 0.24 ^b^
16:1	1.43 ± 0.08	1.37 ± 0.14
18:1	23.98 ± 0.27 ^a^	20.66 ± 0.61 ^b^
18:2n-6 (LA)	33.17 ± 1.00 ^a^	24.11 ± 0.75 ^b^
20:4n-6 (ARA)	0.35 ± 0.02 ^b^	0.59 ± 0.03 ^a^
18:3n-3 (ALA)	3.39 ± 0.14 ^b^	15.85 ± 0.74 ^a^
20:5n-3 (EPA)	0.31 ± 0.03 ^b^	0.68 ± 0.04 ^a^
22:6n-3 (DHA)	4.35 ± 0.22 ^b^	7.34 ± 0.50 ^a^
SFA	32.11 ± 0.64 ^a^	26.81 ± 1.09 ^b^
MUFA	26.21 ± 0.79	24.42 ± 0.82
n-6 PUFA	34.57 ± 0.75 ^a^	27.24 ± 0.86 ^b^
n-3 PUFA	7.95 ± 0.28 ^b^	23.23 ± 0.36 ^a^
n-6 LC-PUFA	0.78 ± 0.02	0.74 ± 0.02
n-3 LC-PUFA	5.72 ± 0.82 ^b^	9.08 ± 1.22 ^a^
PL		
16:0	18.83 ± 0.87 ^a^	13.70 ± 0.76 ^b^
18:0	5.26 ± 0.63 ^a^	3.73 ± 0.24 ^b^
16:1	0.40 ± 0.02	0.42 ± 0.04
18:1	17.06 ± 0.49	15.91 ± 0.31
18:2n-6 (LA)	16.82 ± 0.69 ^a^	8.46 ± 0.64 ^b^
20:4n-6 (ARA)	1.79 ± 0.07 ^b^	2.06 ± 0.14 ^a^
18:3n-3 (ALA)	1.33 ± 0.05 ^b^	4.95 ± 0.05 ^a^
20:5n-3 (EPA)	1.55 ± 0.04	1.90 ± 0.07
22:6n-3 (DHA)	13.70 ± 0.26 ^b^	18.20 ± 0.49 ^a^
SFA	30.08 ± 1.87 ^a^	22.07 ± 1.79 ^b^
MUFA	18.44 ± 0.55	17.39 ± 0.49
n-6 PUFA	20.90 ± 0.73 ^a^	14.68 ± 0.86 ^b^
n-3 PUFA	19.96 ± 0.28 ^b^	25.64 ± 0.59 ^a^
n-6 LC-PUFA	3.26 ± 0.34	3.49 ± 0.47
n-3 LC-PUFA	15.85 ± 0.26 ^b^	21.20 ± 0.50 ^a^

Notes: Values are means ± SE (n = 3) with nine fish per treatment. Values in the same row not sharing a common letter are significantly different (*p* < 0.05).

## Data Availability

The original contributions presented in the study are included in the article, further inquiries can be directed to the corresponding authors.

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
