# Peer review of "Improvement in Muscle Fatty Acid Bioavailability and Volatile Flavor in Tilapia by Dietary α-Linolenic Acid Nutrition Strategy"

_foods, 2024, doi:10.3390/foods13071005_

Round 1

Reviewer 1 Report

Comments and Suggestions for Authors

The study aimed to assess the impact of dietary fatty acid strategies on muscle quality in farmed tilapia. Two diets differing in α-linolenic acid (ALA) content were fed to juveniles over ten weeks. Diet BO (ALA Rich) increased levels of volatile aldehydes and alcohol in muscle tissue and enhanced the expression of genes and protein activities related to lipid mobilization and oxidation.

MS is very interesting, well written and add knowlegde to the field. Some aspects should be improved:

-improve description of the experimental design: replicates, tanks, etc.

- justify level of inclusión: why 0.58 or 1.35%?

- discuss about putative impact on sensorial analysis and the need to contrast analytical results to consumer panel.

. correct letter size in all text

Author Response

  1. Improve description of the experimental design: replicates, tanks, etc.

Response: Thanks for your suggestion, the description of the experimental design was revised in the Section 2.1: In the feeding trial, 120 tilapia GIFI (initial body weight of 170 g) were randomly distributed into 6 tanks (300 L, 20 fish per tank and each dietary group with three tanks), and cultured for 10 weeks with the SO or BO diets. The detailed management of fish husbandry was described in our previous study (Huang et al., 2022).

Huang, X.; Chen, F.; Guan, J.; Xu, C.; Li, Y.; Xie, D. Beneficial effects of re-feeding high alpha-linolenic acid diets on the muscle quality, cold temperature and disease resistance of tilapia. Fish Shellfish Immun. 2022, 126, 303–310

 Justify level of inclusion: why 0.58 or 1.35%?

Response: The present study is a continuation of our previous work (Xie et al., 2022; Huang et al., 2022). The results of the two previous studies demonstrated that tilapia fed with high ALA diets (1.35 %) exhibit better growth and muscle texture compared with these of fish fed with low ALA diets (0.58 %). Therefore, we choose the ALA inclusion level at 0.58% and 1.35% to further analyze the muscle quality in present work.

Xie, D.; Guan, J.; Huang, X.; Xu, C.; Pan, Q.; Li, Y. Tilapia can be a beneficial n-3 LC-PUFA source due to its high biosynthetic capacity in the liver and intestine. J. Agr. Food Chem. 2022, 70(8), 2701–2711.

  1. Discuss about putative impact on sensorial analysisand the need to contrast analytical results to consumer panel.

Response: Thanks for your suggestion, the potential effects of dietary ALA on the sensory quality of tilapia, and the consumer panel deserve to be further investigated. The corresponding discussions were added.

  1. Correct letter size in all text.

Response: Revisions have been made according to your suggestion. please see the revised MS for details.

Reviewer 2 Report

Comments and Suggestions for Authors

The present study assessed the beneficial effects of dietary ALA in Nile tilapia. the study is interesting and valuable. however, the manuscript needs extensive English editing. The sentences are difficult to read.

What are SO, BO?

The first five lines of the abstract are unclear, they should be rewritten.

ATG? DEFINE

The design of the experiment is not clear in the abstract. The authors only mentioned the parameters evaluated.

Define all abbreviations in the abstract  and at first mention.

Introduction: the authors should mention the requirements of Nile tilapia from ALA.

M&M

“muscle samples from our previous feeding trial were used [12].” How did the authors use these samples after this long time? The aforementioned paper was published in 2022, more than two years ago. Moreover, the design for this study is different from the previously published. The authors should clarify.

The rearing system and management of tilapia should be detailed.

5000 ×g

What is the source of A-linolenic acid in the diets? It should be clarified.

Why did the authors use these percentages of ALA?

The experimental design should be mentioned in detail and the authors should mention the management and rearing system of fish.

If the growth performance and proximate composition of whole fish were performed in a previous study, why were they repeated here? All tests performed before in a previous study should be deleted and not repeated. The authors should mention only the design and measures conducted for this study. Any repetition is not accepted.

Figure 1 and Table S3 show the same results. One of them is enough.

Comments on the Quality of English Language

difficult to read

Author Response

1. The present study assessed the beneficial effects of dietary ALA in Nile tilapia. the study is interesting and valuable. however, the manuscript needs extensive English editing.

Response: Thanks for your suggestion. The manuscript has been edited and revised by professor Jeffrey Wragg from University of Charleston. The acknowledgement for his contributions to this article was added in the revised MS.

 2. The sentences are difficult to read. What are SO, BO? The first five lines of the abstract are unclear, they should be rewritten. ATG? DEFINE The design of the experiment is not clear in the abstract. The authors only mentioned the parameters evaluated. Define all abbreviations in the abstract and at first mention.

Response: Thanks for your suggestion, revisions have been made according to your suggestion. Please see the revised MS for details.

 3. Introduction: the authors should mention the requirements of Nile tilapia from ALA.

Response: Revision has been made according to your suggestion. Please see the Introduction section of the revised MS for details.

4. M&M“muscle samples from our previous feeding trial were used [12].” How did the authors use these samples after this long time? The aforementioned paper was published in 2022, more than two years ago. Moreover, the design for this study is different from the previously published. The authors should clarify. The rearing system and management of tilapia should be detailed.

Response: The present work is a continuation of our previous works (Xie et al., 2022; Huang et al., 2022). The muscle samples used in this study were from the previous feeding trial, and the feeding trial was finished at 1/2020 (Huang et al., 2022). In the next 10 months (2/2020-12/2020), the analysis of muscle quality (amino acid and fatty acid compositions, as well as volatile flavor compounds), enzymatic activities and qPCR were conducted. However, the corresponding statistical data were not written into a manuscript in time, and the corresponding manuscript has undergone many revisions too.

The experimental design was revised in the Section 2.1: In the feeding trial, 120 tilapia GIFI (initial body weight of 170 g) were randomly distributed into 6 tanks (300 L, 20 fish per tank, and each dietary group with three tanks), and cultured for 10 weeks with the SO or BO diets. The detailed management of fish husbandry was described in our previous study (Huang et al., 2022).

Xie, D.; Guan, J.; Huang, X.; Xu, C.; Pan, Q.; Li, Y. Tilapia can be a beneficial n-3 LC-PUFA source due to its high biosynthetic capacity in the liver and intestine. J. Agr. Food Chem. 2022, 70(8), 2701–2711.

Huang, X.; Chen, F.; Guan, J.; Xu, C.; Li, Y.; Xie, D. Beneficial effects of re-feeding high alpha-linolenic acid diets on the muscle quality, cold temperature and disease resistance of tilapia. Fish Shellfish Immun. 2022, 126, 303–310.

5. What is the source of A-linolenic acid in the diets? It should be clarified.

Response: Dietary α-linolenic acid (ALA) is from the lipid sources. The soybean oil and linseed oil used in this study contain 6.50 % and 47.44 % ALA, respectively, which were presented detailedly in our previous studies (Xie et al., 2022; Huang et al., 2022). The detailed fatty acid compositions of dietary lipid sources were added in Table S1.

6. Why did the authors use these percentages of ALA?

Response: As mentioned above, the present study is a continuation of our previous work (Xie et al., 2022; Huang et al., 2022). The results of the two previous studies demonstrated that tilapia fed with high ALA diets (1.35 %) exhibit better growth and muscle texture compared with those of fish fed with low ALA diets (0.58 %). Therefore, we choose the ALA inclusion level at 0.58% and 1.35% to further analyze the muscle quality in the present work.

7. The experimental design should be mentioned in detail and the authors should mention the management and rearing system of fish.

Response: Thanks for your suggestion, this question was replied in the question 4 of review 2, please check it.

8. If the growth performance and proximate composition of whole fish were performed in a previous study, why were they repeated here? All tests performed before in a previous study should be deleted and not repeated. The authors should mention only the design and measures conducted for this study. Any repetition is not accepted.

Response: Thanks for your suggestion. Actually, the data of growth performance and proximate composition of whole fish were not presented in this study. The effects of dietary ALA on the growth performance and proximate composition of whole fish were simply touched upon here. Accordingly, the section 3.1 was deleted in the revised MS.

9. Figure 1 and Table S3 show the same results. One of them is enough.

Response: Yes, figure 1 and table S3 came from the same data. Only the figure 1 was displayed in the text of MS, and table S3 was simply submitted as a attachment, which will not be presented in the text of MS.

Round 2

Reviewer 2 Report

Comments and Suggestions for Authors

No further comments